# Apple Watch 6 vs. Galaxy Watch 4: A Validity Study of Step-Count Estimation in Daily Activities

**DOI:** 10.3390/s24144658

**Published:** 2024-07-18

**Authors:** Kyu-Ri Hong, In-Whi Hwang, Ho-Jun Kim, Seo-Hyung Yang, Jung-Min Lee

**Affiliations:** 1Department of Physical Education, Graduate School of Education, Kyung Hee University, Yongin-si 17014, Republic of Korea; rbfl2526@khu.ac.kr; 2Department of Sports Medicine and Science, Graduate School of Physical Education, Kyung Hee University, Yongin-si 17014, Republic of Korea; inn95313@khu.ac.kr; 3Department of Physical Education, College of Physical Education, Kyung Hee University, Yongin-si 17014, Republic of Korea; wnsghrla1@khu.ac.kr; 4School of Global Sport Studies, Korea University, 2511 Sejong-ro, Sejong City 30019, Republic of Korea; trusting2000@naver.com; 5Sports Science Research Center, Global Campus, Kyung Hee University, Yongin-si 17014, Republic of Korea; 6Department of Physical Education, Kyung Hee University, Yongin-si 17014, Republic of Korea

**Keywords:** ActivPAL, Apple Watch 6, Galaxy Watch 4, wearable devices, step counts

## Abstract

The purpose of this study was to examine the validity of two wearable smartwatches (the Apple Watch 6 (AW) and the Galaxy Watch 4 (GW)) and smartphone applications (Apple Health for iPhone mobiles and Samsung Health for Android mobiles) for estimating step counts in daily life. A total of 104 healthy adults (36 AW, 25 GW, and 43 smartphone application users) were engaged in daily activities for 24 h while wearing an ActivPAL accelerometer on the thigh and a smartwatch on the wrist. The validities of the smartwatch and smartphone estimates of step counts were evaluated relative to criterion values obtained from an ActivPAL accelerometer. The strongest relationship between the ActivPAL accelerometer and the devices was found for the AW (r = 0.99, *p* < 0.001), followed by the GW (r = 0.82, *p* < 0.001), and the smartphone applications (r = 0.93, *p* < 0.001). For overall group comparisons, the MAPE (Mean Absolute Percentage Error) values (computed as the average absolute value of the group-level errors) were 6.4%, 10.5%, and 29.6% for the AW, GW, and smartphone applications, respectively. The results of the present study indicate that the AW and GW showed strong validity in measuring steps, while the smartphone applications did not provide reliable step counts in free-living conditions.

## 1. Introduction

Engaging in regular physical activity (PA) is a widely recognized foundational determinant for enhancing overall health [1]. It plays a pivotal role in disease prevention and the promotion of well-being across all age groups [2]. Despite the clear benefits of PA, a 2020 report issued by the World Health Organization (WHO) highlights a significant deficiency in meeting recommended PA guidelines, with a substantial segment of the population falling short of the minimum advised levels of moderate to vigorous PA [3]. Specifically, this deficiency pertains to individuals who do not attain a minimum of 150 min of moderate intensity PA or 75 min of vigorous intensity PA per week. The insufficiency in PA levels may constitute a contributory element to the prevalence of obesity [4] and a spectrum of associated diseases. Consequently, the waning prevalence of PA among adults exerts a pervasive influence on individuals throughout their lifespan and imposes substantial financial burdens on the healthcare system and society at large [5].

Smartphones were once pivotal tools for boosting regular PA through their various applications that measured activity levels. However, their accuracy did face some constraints based on whether they involved direct contact or were non-contact [6]. Both Samsung and iPhone, for instance, featured built-in applications that tracked daily step counts. As technology advanced, smart wearables subsequently emerged, setting a new standard in personal health tracking. This category encompasses a spectrum of devices, including fitness trackers, smartwatches, heart rate monitors, and GPS tracking devices, which have revolutionized our approach to PA and holistic health management [7,8]. Among these technological innovations, smartwatches have emerged as pioneering instruments for the enhancement of individual health. Going beyond their original functions of timekeeping and notification delivery, smartwatches have evolved into versatile health management tools [2,9]. Equipped with an array of sensors, smartwatches enable users to engage in self-monitoring of various health parameters. Self-monitoring of PA stands as an efficacious strategy for improving PA levels [10,11]. Basic fitness tracking capabilities encompass the recording of steps taken, distance covered, and calories expended. Furthermore, contemporary smartwatches extend their capabilities to encompass more sophisticated health metrics, including heart rate, sleep quality, stress levels, and even oxygen saturation.

In the realm of wearable technology, the landscape is dominated by two prominent providers, namely Apple and Samsung. As a titan of the technology industry, Apple’s market value holds the top position, while Samsung is ranked eighth [12]. According to a 2024 study by Counterpoint Research, Apple is the leading company in the smartwatch industry in terms of shipment share, accounting for 26 percent of worldwide shipments as of the first quarter of 2023. This was followed by Samsung with nine percent of the market share [13]. Moreover, since its launch in 2015, the Apple Watch has become the best-selling wearable/smartwatch worldwide, with sales surpassing 12 million units. Samsung has also seen an increasingly positive consumer response, recording second place [14]. With the widespread use of smartwatches, the scale of interventions for PA is also increasing [15]. These two formidable smartwatch offerings have exerted a progressively substantial influence across various domains, with a particular emphasis on their impact in the arena of health monitoring and management [9,16]. However, despite the considerable prevalence of these devices, the accuracy of step-count measurement in free-living conditions remains an underexplored area in the existing body of research.

The quantification of step counts serves as a fundamental and indispensable metric in assessing an individual’s daily PA level. These step counts, conventionally monitored through devices such as pedometers and contemporary smartwatches, provide a cumulative measure of daily energy expenditure. There is a close and intricate relationship between step counts, the volume of PA, and its associated intensity [9,17]. Therefore, this study investigated step counts, one of the fundamental metrics of evaluating an individual’s PA level in daily life. Recent research conducted by Chaudhry (2020) has elucidated that the utilization of pedometers or similar step-count tracking devices specifically designed for health monitoring leads to a notable improvements in recorded step counts compared to conventional care groups [18,19]. This improvement is made possible by the sensors embedded within modern smartphones and smartwatches, which are capable of detecting and recording human activities [20]. These sensors, which include tri-axis acceleration and tri-axis angular velocity measurements, are critical for the accurate quantification of an individual’s step count [21].

In the current global context of increased smartwatch and smartphone usage and heightened interest in personalized health monitoring and management, it is essential to evaluate the validity of PA measurements by these devices. Previous research indicates that while smartwatches accurately monitor heart rate during activities like walking and running, their accuracy declines with increased intensity of activity [22,23]. Despite several studies addressing the accuracy of activity measurement by smartwatches, a comprehensive examination of step-count accuracy in free-living conditions remains lacking. Specifically, comparative studies on the accuracy of major smartwatches, such as the Apple Watch and Samsung Galaxy Watch, are scarce. Additionally, there is limited analysis of accuracy variations across different activity levels, which is essential for evaluating the practical utility and validity of wearable devices. Therefore, this study aims to assess the accuracy of step counts measured by the Apple Watch and Samsung Galaxy Watch in daily life, comparing them with the ActivPAL accelerometer as a criterion measure. This research seeks to provide an in-depth evaluation of the practical application and contribution of smartwatches to health monitoring.

## 2. Materials and Methods

### 2.1. Participants

In this study, a total of 104 participants, comprising 36 Apple Watch 6 users (17 males and 19 females), 25 Galaxy Watch 4 users (10 males and 15 females), and 43 individuals (21 males and 22 females) who did not wear smartwatches (referred to as the “Smartphone Application” group), were included. The number of participants required to ensure sufficient statistical power was calculated using G*Power 3.1.9.7 software and the effect size reported in previous studies [24]. G*Power is a program used to calculate the necessary sample size before conducting a study. Based on an effect size (Cohen’s f) of 0.25, a Type I error rate of 5% (α error), and a power of 95% (power = 1 − β), the required total sample size was determined to be 45 participants. The participants, aged between 18 and 53 years, were recruited through word of mouth and direct approach. This study excluded variables that significantly affect PA in daily life, such as chronic diseases, physical disabilities, and pulmonary conditions, ensuring the selection of participants who could wear the devices without difficulty. In addition, if participants did not wear their smartwatches for more than 2 h a day, excluding sleep time, or if there was a discrepancy in the wearing periods between their ActivPAL devices and their smartwatches, the data were considered missing and excluded. To minimize any unreasonable discrepancies in measurements, participants were restricted from engaging in high-intensity activities and were instructed to record the time if they engaged in such activities. Each participant was instructed to wear both a smartwatch on their wrist and an ActivPAL device on their thigh continuously for 24 h. Furthermore, the smartphone application group consisted of participants who solely wore the ActivPAL device on their thighs. The number of steps of the participants in this group was measured by utilizing smartphone applications in conjunction with the ActivPAL device. To minimize error and ensure precise measurement, participants were instructed to continuously wear their smartwatches throughout each day, excluding sleep time, and to carry their mobile phones with them at all times. The Institutional Review Board of Kyung Hee University approved the study (KHGIRB-23-095).

### 2.2. Instruments

The study’s participants were instructed to utilize two distinct categories of activity monitoring devices. Each participant was required to wear a smartwatch on their wrist (Figure 1) in addition to wearing an ActivPAL device affixed to their thigh (Figure 2). In the case of the smartphone applications, they were held in the hand or carried in the participant’s clothes, while the smartphone-based accelerometer measured the step counts in the direction of the starting point of motion for each step [25]. Table 1 presents a comprehensive list of the devices used in this study and some extra information.

#### 2.2.1. ActivPAL (ActivPAL 4 v8.12.6) (AP)

The ActivPAL accelerometer is a small capacitance-based accelerometer (5.3 cm × 3.5 cm × 0.7 cm) and light (15 g). During the study, the ActivPAL device was affixed to the thigh using hypoallergenic tape and was not detached except in special or hazardous situations. It could later be verified through the data analysis process. This procedure minimized errors in the criterion values. The ActivPAL device functions as an instrument for analyzing PA based on body positioning or morphology. Precisely, it possesses the capability to discern between three distinct activity states: sitting/lying, standing, and stepping, as expounded upon in reference [26]. The ActivPAL accelerometer was selected as the criterion measure for this study due to its proven accuracy and reliability in measuring PA and stepping activities [27,28]. The ActivPAL accelerometer is widely recognized in the field of PA research for its high validity in various settings, including free-living conditions. Additionally, it has been extensively validated against other gold-standard methods, making it an ideal benchmark for evaluating the performance of consumer-grade wearable devices such as smartwatches.

#### 2.2.2. Smartwatches (Apple Watch 6 and Galaxy Watch 4)

In this study, we evaluated the functionality of two distinct smart wearables devices: the Apple Watch 6 and the Galaxy Watch 4, occupying the highest ranks in the current technology market values [12]. This study selected the latest versions available at the time of the research to investigate the accuracy of their measurements. The Apple Watch Series 6 and the Galaxy Watch 4 are equipped with an array of sensors designed to monitor various health metrics. The Apple Watch Series 6 includes an accelerometer, gyroscope, heart rate monitor, and blood oxygen (SpO_2_) sensor. These sensors enable the device to track physical activity, detect irregular heart rhythms, and measure blood oxygen levels [29]. The Galaxy Watch 4 also features an accelerometer, gyroscope, heart rate monitor, and body composition sensor, which provides additional insights into the user’s health by estimating metrics such as body fat percentage and skeletal muscle mass. The accelerometer and gyroscope in both devices allow for the detection of movement and orientation, which are crucial for accurately counting steps and assessing physical activity levels. The heart rate monitors use photoplethysmography (PPG) to measure the user’s pulse by detecting blood flow changes in the wrist. The SpO_2_ sensor in the Apple Watch and the body composition sensor in the Galaxy Watch enhance the devices’ abilities to provide comprehensive health monitoring. By incorporating these advanced sensors, both smartwatches offer robust health tracking capabilities that are critical for the accurate measurement of step counts and other physical activity metrics. This could provide valuable information to a wide range of users. These devices are seamlessly integrated with their respective smartphones, the iPhone and Samsung Galaxy phone, facilitating the effortless visualization of data through mobile applications.

Both wearable devices are equipped with the capability to accurately quantify step counts, relying on tri-axis acceleration and tri-axis angular velocity measurements [30] to discern various physical activities, including walking and running. This capacity enables the smartwatches to effectively monitor and record step counts by analyzing the user’s PA. Furthermore, wearable sensor technology advancements are closely linked to the development of low-power soft transistors and flexible electronic devices. Liu et al. (2024) demonstrated that low-power soft transistors could greatly enhance the efficiency of wearable devices by reducing energy consumption and maintaining stable performance over extended periods [31]. Additionally, Zhang et al. (2023) explored how flexible electronic devices could be utilized in cardiovascular healthcare monitoring, explaining that the application of flexible electronics plays a crucial role in improving the reliability and accuracy of wearable devices [32]. Thus, the accurate step-count measurement capabilities of the Apple Watch 6 and Galaxy Watch 4 are further enhanced by the application of the latest low-power soft transistor and flexible electronics technologies. The collected step data are automatically synchronized with the corresponding mobile application; namely, Apple Watch’s Health application, and Galaxy Watch’s Samsung Health application. All study participants were instructed to wear these smartwatches continuously for each 24 h period, with exceptions made solely in cases of health-related emergencies.

### 2.3. Study Protocol

At the beginning of the data acquisition, anthropometric measurements were ascertained. Participants’ height and weight were garnered through self-report health history, subsequently permitting the calculation of their Body Mass Index (BMI) as the ratio of weight (kg) to height squared (m^2^). Subsequently, the ActivPAL device was affixed to the participants’ thighs as per the manufacturer’s recommendation. While both wearing the ActivPAL device and either wearing a smartwatch or carrying a smartphone featuring a health application, participants’ step counts were recorded in free-living conditions for 24 h. During a follow-up meeting, the ActivPAL device and the amassed data were retrieved. To enhance the precision and reliability of the study’s outcomes, an assessment was conducted to identify instances where the smartwatch had been detached for more than one hour, and such instances were categorized as missing data values.

### 2.4. Data Analysis

All data processing and statistical analyses in this study were conducted using SPSS version 28.0 (SPSS Inc., Chicago, IL, USA). Descriptive statistics were employed to succinctly summarize the demographic characteristics of the study participants, including gender, age, height, weight, and BMI. To evaluate the correlation between the smartwatch (or smartphone application) and the ActivPAL accelerometer in measuring step counts, Kendall’s tau-b and Pearson’s correlation coefficients were computed. The Mean Absolute Percent Error (MAPE) was calculated as the average of the absolute difference between the criterion measure value (as measured by the ActivPAL accelerometer) and the values recorded by each device, divided by the criterion measure value, then multiplied by 100. Furthermore, a paired-sample *t*-test was utilized to identify any significant mean differences between the ActivPAL accelerometer and each respective device. In analyzing the step-count data from all the devices, we applied equivalence testing, a novel statistical approach designed to assess the equivalence between diverse measures, as opposed to solely testing against a null hypothesis of zero difference [33]. To examine the proportional systematic biases, Bland–Altman plots were utilized. These plots incorporate 95% limits of agreement and fitted lines derived from regression analyses comparing the criterion measure to the differences in values obtained from the ActivPAL accelerometer and each other device. Parameters such as intercept and slope were included in the analyses to provide a comprehensive evaluation of the agreement between the devices.

## 3. Results

A total of 104 participants (48 male and 56 female) were involved in this study. All participants had smartwatches or smartphones. Table 2 presents comprehensive descriptive statistics summarizing the anthropometric characteristics of all the study participants.

Meanwhile, Table 3 displays the correlation coefficients (r) representing the relationships between the ActivPAL accelerometer and three distinct devices; the Apple Watch 6, Galaxy Watch 4, and smartphone applications. These devices exhibited substantial correlations with the ActivPAL device (Apple Watch 6: r = 0.986, *p* < 0.01; Galaxy Watch 4: r = 0.824, *p* < 0.01; and smartphone applications: r = 0.926, *p* < 0.01), with the strongest association observed between the ActivPAL device and the Apple Watch 6.

Figure 3 shows the MAPE for each device, computed as the average absolute error relative to the ActivPAL criterion. The smallest errors were associated with the Apple Watch 6 (6.4%), followed by the Galaxy Watch 4 (10.5%), and the smartphone applications (29.6%). These findings underscore the superior accuracy of smartwatches in comparison to smartphone applications, with the Apple Watch 6 demonstrating the lowest MAPE and, consequently, the highest accuracy.

The results from equivalence testing are delineated in Figure 4, demonstrating the substantial equivalence of the smartwatches with the criterion measure. Specifically, the step counts recorded by the Apple Watch 6 and the Galaxy Watch 4 largely fell within the equivalence zone (90% confidence interval within the equivalence zone: Apple Watch 6 = 7770.0–9886.1 and Galaxy Watch 4 = 7732.6–10022.9). Conversely, the step counts from the smartphone applications were outside the equivalence zone (smartphone applications’ 90% confidence interval outside of the equivalence zone: 6681.0–9718.2).

Bland–Altman plot analyses, as displayed in Figure 5, were carried out to assess the distribution of errors and identify the potential proportional systematic biases in the estimates. These plots show the differences between the criterion measure (the ActivPAL device) and each device on the *Y*-axis, plotted against the average step counts of the criterion and each device on the *X*-axis. Notably, the narrowest 95% limits of agreement were observed for the Apple Watch 6 (difference = 2494.7), with slightly wider limits for the Galaxy Watch 4 (difference = 3609.7). In contrast, the smartphone applications exhibited the widest limits (difference = 9455.9). The distribution of points around the mean was more concentrated for the Apple Watch 6, followed by the Galaxy Watch 4, while the smartphone applications displayed more dispersed data points. Furthermore, the fitted line slopes for all the devices were found to be statistically nonsignificant (Apple Watch 6: slope = −0.10, *p* = 0.923; Galaxy Watch 4: slope = −0.11, *p* = 0.891; and smartphone applications: slope = 0.31, *p* = 0.442), indicating the absence of substantial patterns of proportional systematic bias with these devices.

Figure 6 illustrates the outcomes of a paired-sample *t*-test comparing the criterion measure (the ActivPAL accelerometer) against the various devices. This analysis indicates a lack of statistically significant difference between the Galaxy Watch 4 and the criterion measure (*p* = 0.198), with a small effect size (d = 0.265). However, notable disparities were observed between the criterion measure and the Apple Watch 6 (*p* < 0.05, *p* = 0.033), reflecting a small effect size (d = 0.369), as well as with the smartphone applications (*p* < 0.01), with a large effect size (d = 0.945).

## 4. Discussion

This is a valuable study to assess the validity of the Apple Watch, the Galaxy Watch, and smartphone applications for measuring step counts compared to the criterion measure of the ActivPAL accelerometer under free-living conditions. Previous studies have also documented the measurement accuracy of the smartwatches, particularly in the reliable quantification of heart rate and energy expenditure [34,35], and it is evident that the measurement accuracy of the smartwatches varies with the intensity of the exercise [22,23]. Additionally, according to prior studies that measured the Apple Watch’s step counts on a treadmill at varying speeds, the most accurate readings occurred at moderate pace, with the tendency to undercount or overcount at faster or slower speeds, respectively [36]. However, questions remain regarding the details of how the Apple Watch and the Galaxy Watch measure PA [15]. Furthermore, research is required related to measuring PA in free-living conditions, not fixed to any specific intensity. Therefore, the present study assists smartwatch users in accurately gauging their PA levels in daily life, and the significance lies in the ability to increase exercise and enhance overall health based on such precise data.

Numerous studies have been conducted to assess variations in measurement accuracy associated with different types of research-oriented activity monitors [37,38,39]. Hergenroeder et al. conducted a study wherein participants underwent two 100-step walking tests while simultaneously wearing three research-grade activity monitors. The step-count accuracy of these devices was evaluated by comparison with a manual step-count hand tally [38]. Their findings indicated that the ActivPAL accelerometer, when positioned on the thigh, consistently exhibited a high level of accuracy, accounting for 97.3 ± 11.1% of manually tallied steps; in contrast to the ActiGraph GT9X, which, when worn at the waist, exhibited a considerably wider range of measurement variance (51.4 ± 35.7%) in relation to the criterion measure. Additionally, the study highlighted that the ActivPAL accelerometer provided a reasonable degree of accuracy in measuring gait speed, with values ranging from 86.8 ± 14.0% to 95.1 ± 9.2% [38]. This study’s findings are consistent with the body of literature supporting the accuracy of the ActivPAL accelerometer. In a study by An et al. titled “Accuracy of Inclinometer Functions of the ActivPAL and ActiGraph GT3X+: A Focus on Physical Activity”, the ActivPAL device was shown to exhibit high accuracy for measuring stepping and it proved to be significantly accurate in detecting various forms of PA, including sitting, standing, and stepping [27]. In the current study, the ActivPAL device served as the criterion measure, reinforcing the study’s robust validity.

Comparative analyses involving smartwatches, namely the Apple Watch and Galaxy Watch, exhibit a notable degree of concordance with the criterion measure, accompanied by low MAPE values. Specifically, step-count data originating from the Apple Watch demonstrate a statistically significant equivalence with the criterion measure and an MAPE below 10%, signifying a high level of accuracy, which is consistent with prior investigations suggesting that the Apple Watch boasts superior accuracy compared to the established criteria [40,41]. Notably, MAPE values below the 10% threshold, as recorded by the Apple Watch, are indicative of a robust level of accuracy [42]. Furthermore, the Apple Watch exhibits a statistically significant correlation with the criterion measure (*p* < 0.01). While the Apple Watch demonstrates the most favorable overall performance, it is noteworthy that the Galaxy Watch yields commendable results within this study. The Galaxy Watch displays a statistically significant correlation with the criterion measure (*p* < 0.01). It records an MAPE value of 10.5%, a figure closely approaching the 10% threshold, indicative of a high degree of accuracy [42]. It is worth mentioning that only the Galaxy Watch exhibited no significant difference compared to the criterion measure.

Ultimately, the smartphone applications also showed a pronounced correlation with the ActivPAL device. Existing research indicates that, regardless of the smartphone’s position, whether it is in a bag or a pants pocket, high accuracy and generally low variability are demonstrated. Additionally, all accelerometer apps tend to systematically underestimate steps during free-living walking activities [43]. However, in this study they yielded notably lower accuracy in measuring the number of steps. Specifically, the MAPE of the smartphone applications showed the highest value at 29.6%, deviating considerably from the comparable range within the equivalence zone. In comparison to the smartwatches this discrepancy was substantial. Previous research corroborates these findings, as exemplified by Presset et al. [44], who reported diminished step-count accuracy of smartphone applications when smartphones were positioned in the most comfortable locations, such as within a “jacket”. Notably, for the Runtastic Pedometer smartphone application, the ‘jacket’ placement ranked as the least accurate among the three positions assessed (i.e., arm, belt, and jacket) [44]. This suggests that smartphones, which must be carried in clothing or bags, could show significant discrepancies in accuracy compared to smartwatches, which are attached to the body. Compared to the previous study, which was conducted under controlled conditions, such as treadmill settings, the current study proceeded within unconstrained free-living conditions, which evidently underscored the diminished accuracy inherent in smartphone applications for step-count quantification.

This is one of the first studies to compare the Apple and Galaxy Watches in free-living conditions. Notably, these smartwatches are globally recognized and extensively utilized for fitness and health tracking due to their acknowledged precision and widespread user acceptance. Furthermore, the current research differs significantly from previous studies, which predominantly compared smartwatches to wrist-worn accelerometers, such as the ActiGraph GT9X and Fitbit, as the criterion measures [2,45]. Instead of these devices, we employed the ActivPAL accelerometer, known for its minimal margin of error in measuring PA. Lastly, our study was conducted in the context of uncontrolled daily life, which was a distinguishing feature. While numerous studies have revealed the accuracy of smartwatches in step counting within controlled environments like laboratories and treadmills, the literature is less populated with studies addressing step counting during daily activities. Given the ubiquity of smartwatch usage in individuals’ everyday lives, our findings assume heightened relevance and applicability. They stand poised to offer valuable insights for self-monitoring of PA and health through smartwatch technology, potentially highlighting meaningful implications for health-conscious individuals seeking to leverage these devices for wellness management. The authors acknowledge several limitations of this study. Firstly, the small sample sizes employed in the research must be recognized. Data collection in an uncontrolled environment resulted in the exclusion of numerous data points, particularly affecting the sample size for the Galaxy Watch, which was less than 30. Despite conducting a normality test, the constrained sample size inevitably restricted the study’s generalizability. The small sample size unavoidably constrained the study. Future research should aim to gather larger and more representative samples to yield more comprehensive outcomes and measure a broader spectrum of physical activities, including factors such as sleep patterns and sedentary behaviors, to enhance the study’s overall robustness and applicability.

## 5. Conclusions

In conclusion, we examined the accuracy of step-counting functionality across the Apple Watch 6, Galaxy Watch 4, and smartphone applications. The results unequivocally establish the Apple Watch 6 as the most accurate among the devices, a distinction supported by statistically significant findings. Furthermore, the Galaxy Watch 4 also demonstrated a level of step-counting accuracy akin to that of the Apple Watch 6. Conversely, smartphone applications lagged in terms of accuracy. Consequently, both the Apple Watch 6 and Galaxy Watch 4 are expected to provide data with sufficient accuracy to allow users to plan, measure, and enhance their PA in everyday life. On the other hand, data provided by smartphone applications demonstrated lower validity and reliability for contributing to PA. These findings underscore the value and validity of smartwatches as effective tools for routine self-monitoring of daily steps and the management of PA.

## Figures and Tables

**Figure 1 sensors-24-04658-f001:**
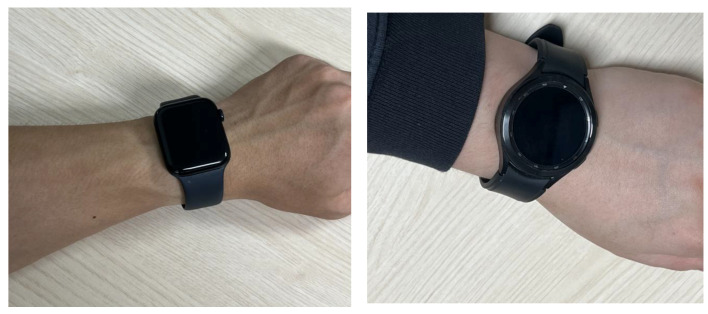
Images showing how the Apple Watch (**Left**) and Galaxy Watch (**Right**) are worn on the wrist.

**Figure 2 sensors-24-04658-f002:**
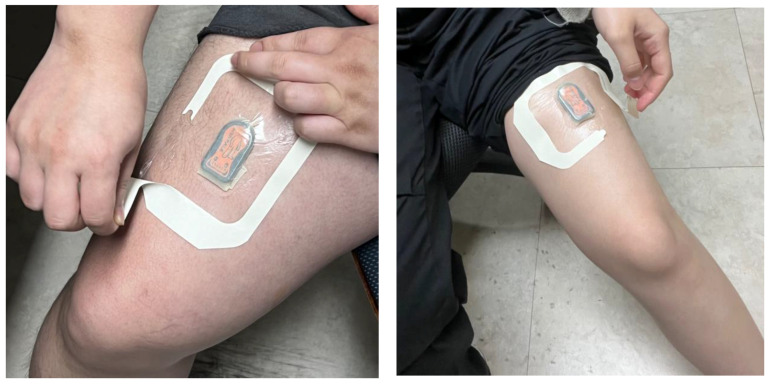
Images showing how the ActivPAL accelerometer is worn on the thigh.

**Figure 3 sensors-24-04658-f003:**
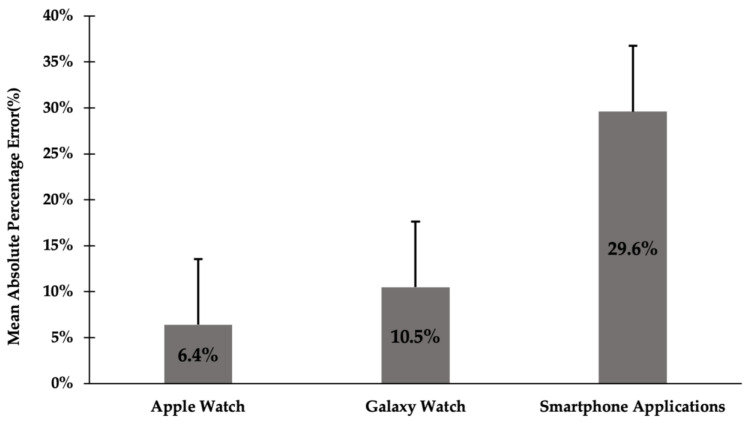
Mean absolute percentage error (±SD) for all devices; Apple Watch, Galaxy Watch, and smartphone applications.

**Figure 4 sensors-24-04658-f004:**
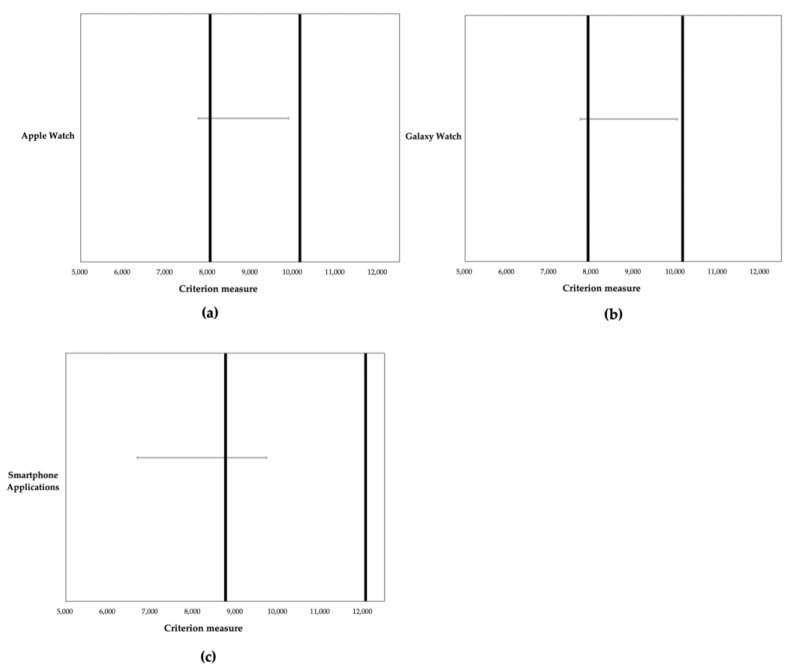
Results from 90% equivalence testing for agreement between the ActivPAL accelerometer and the devices: (**a**) Apple Watch; (**b**) Galaxy Watch; and (**c**) smartphone applications. Dark lines indicate the proposed equivalence zone (±10% of the mean); Grey bars indicate the 90% confidence interval for the means of the devices.

**Figure 5 sensors-24-04658-f005:**
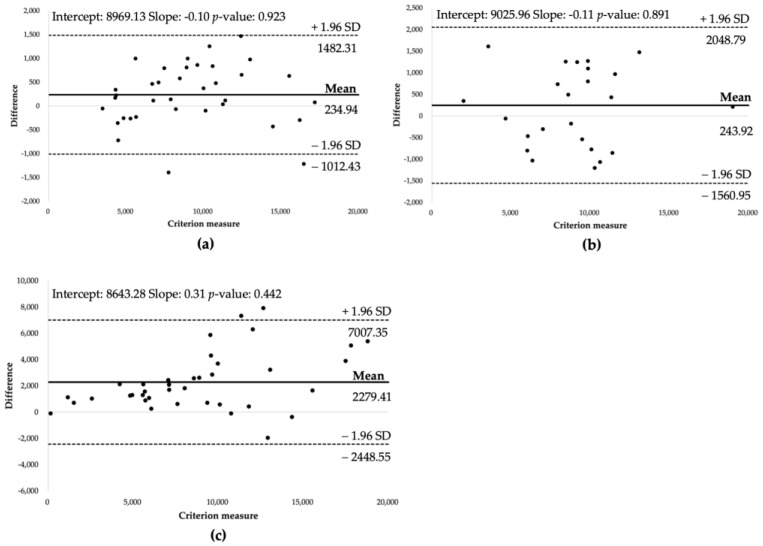
Bland–Altman plots showing the agreement of step counts between ActivPAL accelerometer and each device: (**a**) Apple Watch; (**b**) Galaxy Watch; and (**c**) smartphone applications. Dots represent the difference between the step count measured by each device and the criterion value measured by ActivPAL.

**Figure 6 sensors-24-04658-f006:**
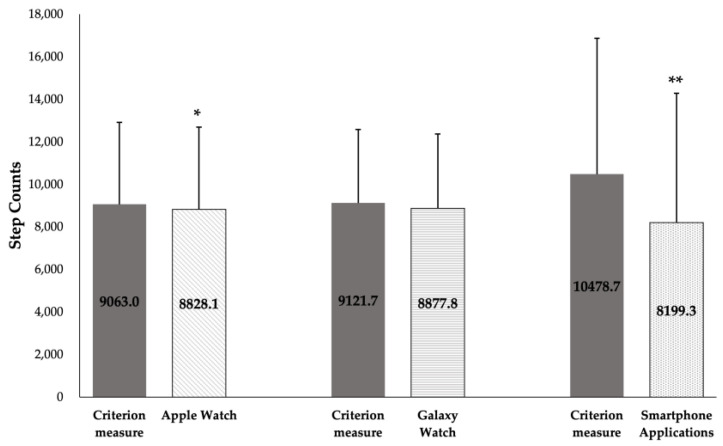
Paired-sample *t*-test (±SD) between criterion measure (the ActivPAL accelerometer) and each device; Apple Watch, Galaxy Watch, and smartphone applications. ** p*-value < 0.05, ** *p*-value < 0.01.

**Table 1 sensors-24-04658-t001:** A list of devices included in this study by versions used, locations worn, and software.

Device	Version	Location Worn	Software
ActivPAL (PAL Technologies Ltd., Glasgow, UK)	ActivPAL4	Thigh	AcivPAL4 v8.12.6
Apple Watch (Apple, Cupertino, CA, USA)	Apple Watch se2	Wrist	Apple Watch 6 v9.3.1
Galaxy Watch (Samsung, Suwon-si, Republic of Korea)	Galaxy Watch 4	Wrist	Galaxy Watch 4 v2.2.11.23082851
Apple iPhone (Apple, Cupertino, CA, USA)	iPhone	-	iPhone app; Health (iOS 16)
Samsung Galaxy Phone (Samsung, Suwon-si, Republic of Korea)	Galaxy Phone	-	Galaxy Phone app; Samsung Health (UI 5.0)

app: application.

**Table 2 sensors-24-04658-t002:** Physical characteristics of participants (*n* = 104).

Variables			Apple Watch		Galaxy Watch		Smartphone Applications	
			No. (%)	Mean (SD)	No. (%)	Mean (SD)	No. (%)	Mean (SD)
Gender	Male		17 (47.2%)	-	10 (40%)	-	21 (48.8%)	-
	Female		19 (52.8%)	-	15 (60%)	-	22 (51.2%)	-
Anthropometrics	Male	Age (year)	17	25.7 (5.6)	10	31.1 (11.8)	21	27.9 (8.6)
		Height (cm)		175.2 (0.1)		177.0 (0.0)		176.0 (0.1)
		Weight (kg)		73.9 (6.9)		82.7 (13.1)		77.4 (8.7)
		BMI (kg/m^2^)		24.1 (1.6)		26.4 (4.3)		25.0 (2.5)
	Female	Age (year)	19	23.3 (3.6)	15	27.3 (10.2)	22	25.0 (5.1)
		Height (cm)		163.8 (0.0)		165.9 (0.0)		163.8 (0.0)
		Weight (kg)		50.3 (3.6)		52.3 (1.8)		52.6 (3.7)
		BMI (kg/m^2^)		18.8 (1.4)		19.0 (1.3)		19.6 (1.3)
	Total	Age (year)	36	24.4 (4.7)	25	28.8 (10.8)	43	26.4 (7.1)
		Height (cm)		169.2 (0.1)		170.4 (0.1)		169.7 (0.1)
		Weight (kg)		61.5 (13.1)		64.5 (17.2)		64.7 (14.1)
		BMI (kg/m^2^)		21.3 (3.1)		22.0 (4.6)		22.2 (3.4)

BMI: Body Mass Index; SD: standard deviation.

**Table 3 sensors-24-04658-t003:** Correlations between the ActivPAL accelerometer and the Apple Watch, Galaxy Watch, and non-wearable smartphone applications (*n* = 104).

	Correlation			
	ActivPAL	Apple Watch	Galaxy Watch	Smartphone Applications
ActivPAL	1	0.986 **	0.824 **	0.926 **

** Correlation is significant at the 0.01 level (2-tailed).

## Data Availability

The data presented in this study are available on request from the corresponding author.

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
