# Peer review of "Apple Watch 6 vs. Galaxy Watch 4: A Validity Study of Step-Count Estimation in Daily Activities"

_sensors, 2024, doi:10.3390/s24144658_

Round 1
Reviewer 1 Report
Comments and Suggestions for Authors
This study validated the Apple Watch 6 and Galaxy Watch 4 against the ActivPAL accelerometer as gold standard. The manuscript is generally well written and organised, presenting useful information in a logical manner. The findings can inform smart watch users who wish to monitor they physical activities, and can guide developers to improve and innovate on wearable technologies. I have no major concerns on this manuscript; just a few minor comments for the authors to consider:
· MAPE is not defined in the abstract.
· Before stating the study aim, it will be good to explicitly mention the research gap. The novelty of the current work can also be emphasized at the end of the Introduction.
· Why did the author choose ActivPAL accelerometer as a criterion measure?
· Table 1 can be expanded to include pictures of how the device was placed on various part of the body.
· How is the sample size of 104 determined?
· Figure 1 can be improved. The black text is hard to read within the columns. Use of colour columns, and may be adding pictured of the device, may help enhance the visual effect.
· Figure 2 can also be improved with more detailed explanation in the capture or annotation in the graphs. The font size is too small. By reading the figure alone, the readers may not be able to follow the main message this figure intends to illustrate.
· Figure 3 – Please increase the font size and resolution such that the figure is not blurry.
· Figure 4 – Please improve the presentation (use colour?)
· Results – Please report the exact p-values (p = ….) and not just p < 0.05 or p < 0.01.
· Please add the effect size (e.g. Cohen’s d for t-tests) to supplement p-values.
· Discussion – Please proofread the English throughout. There are odd sentences such as “This is the valuable study …”. Seek professional help if necessary.
· “Numerous studies have been conducted …” – Please cite some studies at the end of this sentence to support your statement.
· “Informed consent has been obtained from the patient…” Does this study involve patients or generally healthy participants? Please clarify the study population.
Comments on the Quality of English LanguageSome proofreading (esepcially for the Discussion) will be good.
Author Response
Thank you for the detailed and specific feedback. We have written responses to all the questions in order and uploaded them in the file below. Thank you.

Reviewer 2 Report
Comments and Suggestions for Authors
The manuscript reports a validity study of step count estimation in daily activities. This study is interesting, and the results indicate that smartwatches are effective tools for self-monitoring of daily steps and the management of physical activity. However, there are still some minor issues to be addressed before it can be considered for publication.
1. How is the sample (104 participants) determined? Does the sample size affect the conclusion? It is suggested to increase the data size of this sample and broaden the distribution of different age groups.
2. Besides the free-living conditions, what’s the possible results for different wearable devices under the state of sports? Will it make a big difference?
3. The description of sensors can be strengthened in favor of a better fit with the journal's scope and the research background, for example by citing the reports on wearable sensors, doi.org/10.1016/j.xinn.2024.100616 and doi.org/10.1016/j.xinn.2023.100485.
4. Figure 3 is not clear, which should be improved. Please also improve the quality of other figures. I recommend using color styles for all the images.
Comments on the Quality of English Languageno
Author Response

(The authors gave the same response as above.)

Reviewer 3 Report
Comments and Suggestions for Authors
this topic is very interesting, and has significant practice. however, some revision is necessary.
(1) in the second paragraph, why this article limit the topic in the step counts? it needs a simple explanation
(2) in the last paragraph of the introduction section, the other wearable devices, such as PPG, ECG technology, is not related to this topic;
(3) why choose the ActivPAL as the criteria measure in the experiment;
(4) in the section 2.1, how to just the unreasonable discrepancy between the ActivPAL as the criteria measure and the smartwatch
(5) the different number of males and females for two groups will affect the comparison results? male and female individuals have different physical activity level and habit.
(6) for the individuals, his data recorded by the tested wearables is credible or repeatable?
(7) in figure 3, a big discreteness was observed. how to judge the agreement?
(8) in figure 4, why choose different significant level?
Author Response

(The authors gave the same response as above.)

Round 2
Reviewer 2 Report
Comments and Suggestions for Authors
All the questions/comments listed were fully answered in an evasive or negative manner, and there was no substantive response or improvement, in particular for some key questions. Thus we do not agree to publish it at its current state.
Comments on the Quality of English Languageno
Author Response
Thank you for the specific and meaningful feedback. I sincerely answered the given questions and uploaded them as word files. Thank you.
